# A Comparative Study of the Effects of the 1872 Storm and Coastal Flood Risk Management in Denmark, Germany, and Sweden

Caroline Hallin [1,2,*], Jacobus L. A. Hofstede [3], Grit Martinez [4], Jürgen Jensen [5], Nina Baron [6], Thorsten Heimann [7], Aart Kroon [8], Arne Arns [9], Björn Almström [1], Per Sørensen [10] and Magnus Larson [1]

1    Division of Water Resources Engineering, Lund University, John Ericssons väg 1, 223 63 Lund, Sweden; bjorn.almstrom@tvrl.lth.se (B.A.); magnus.larson@tvrl.lth.se (M.L.)
2    Department of Hydraulic Engineering, Delft University of Technology, Stevinweg 1, 2628 CN Delft, The Netherlands
3    Schleswig-Holstein Ministry of Energy Transition, Agriculture, Environment, Nature and Digitization, Mercatorstrasse 3-5, 24105 Kiel, Germany; Jacobus.Hofstede@melund.landsh.de
4    Ecologic Institute, Pfalzburgerstraße 43-44, 10717 Berlin, Germany; grit.martinez@ecologic.eu
5    Research Institute for Water and Environment, University of Siegen, Paul-Bonatz-Str. 9-11, 57076 Siegen, Germany; juergen.jensen@uni-siegen.de
6    The Emergency and Risk Management Program, University College Copenhagen, Sigurdsgade 26, 2200 Copenhagen, Denmark; NIBA@kp.dk
7    Environmental Policy Research Centre, Freie Universität Berlin, Ihnestraße 22, 14195 Berlin, Germany; t.heimann@fu-berlin.de
8    Department of Geosciences and Natural Resource Management, University of Copenhagen, Øster Voldgade 10, 1350 Copenhagen, Denmark; ak@ign.ku.dk
9    Faculty of Agricultural and Environmental Sciences, University of Rostock, Justus-von-Liebig-Weg 6, 18059 Rostock, Germany; arne.arns@uni-rostock.de
10   Kystdirektoratet, Højbovej 1, 7620 Lemvig, Denmark; pso@kyst.dk
*    Correspondence: caroline.hallin@tvrl.lth.se

**Abstract:** From November 12th to 13th in 1872, an extreme coastal flood event occurred in the south Baltic Sea. An unusual combination of winds created a storm surge reaching up to 3.5 m above mean sea level, which is more than a meter higher than all other observations over the past 200 years. On the Danish, German, and Swedish coasts, about 300 people lost their lives. The consequences of the storm in Denmark and Germany were more severe than in Sweden, with significantly larger destruction and higher numbers of casualties. In Denmark and Germany, the 1872 storm has been more extensively documented and remembered and still influences local and regional risk awareness. A comparative study indicates that the collective memory of the 1872 storm is related to the background knowledge about floods, the damage extent, and the response to the storm. Flood marks and dikes help to remember the events. In general, coastal flood defence is to the largest degree implemented in the affected areas in Germany, followed by Denmark, and is almost absent in Sweden, corresponding to the extent of the collective memory of the 1872 storm. Within the affected countries, there is local variability of flood risk awareness associated with the collective memory of the storm. Also, the economic dependency on flood-prone areas and conflicting interests with the tourism industry have influence on flood protection decisions. The processes of climate change adaptation and implementation of the EU Floods Directive are slowly removing these differences in flood risk management approaches.

**Keywords:** 1872 storm; collective memory; historical storms; flood risk management

## 1. Introduction

The sensitivity for coastal flooding is usually assessed by using statistical analyses of measured water levels and by applying hydrodynamic models. The flooding sensitivity is

then translated to flooding risks, where the value of the flooded area is also incorporated. The next step is to generate risk awareness and management plans to reduce the risks. Kaplan and Garrick [1] stated that risk awareness in itself leads to decreased risks. However, in areas where coastal floods result from rare events not even experienced by all generations, the phenomenon of confirmation bias makes it difficult for people to consider and prepare for extreme flood events [2]. In our personal perception of risk, emotion is often more powerful than reason [2–4]. The connection to the disaster loses its emotional component without personal association, like first-hand experience or recollections of parents and grandparents [5–7]. Then, the collective memory plays an important role in maintaining risk awareness of extreme events that have not been self-experienced [8–10].

In 1872, a disastrous storm event affected the countries bordering the western parts of the south Baltic Sea. At the Danish, German, and Swedish coasts, around 300 people died, and more than 15,000 people lost their homes in the German state Schleswig-Holstein alone [11,12]. The water level variations in the south Baltic Sea are usually affected by modest surges due to wind setup. However, during the 1872 storm, the water levels in, e.g., Travemünde, Germany, locally rose to 3.3 m above normal sea level at that time [13,14]. This record-breaking event was more than a meter higher than all other events that have been observed at this station since it has started operating in 1826 (Figure 1).

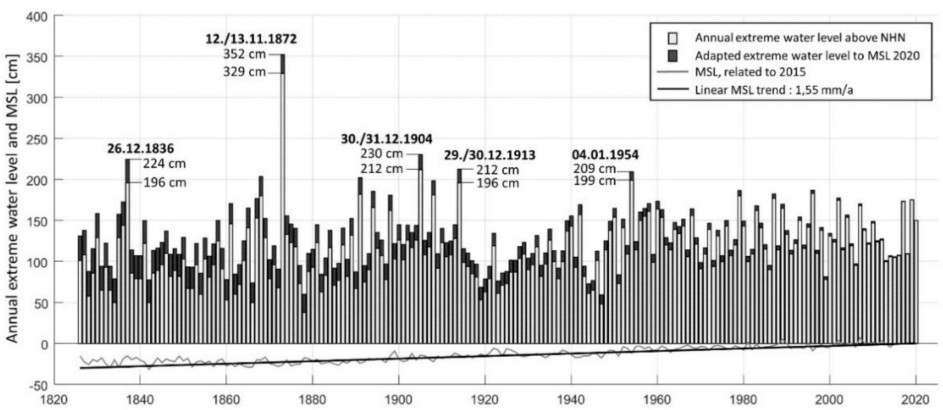

**Figure 1.** Annual extreme water level above NHN (German reference datum), annual extreme water levels adopted to MSL in 2020, MSL, and MSL trend at Lübeck-Travemünde gauge station for the period 1826 to 2020. The figure is based on data from Jensen and Töppe, and Kelln [14–16].

Historically, the worst experienced disaster was commonly used as design criteria for flood protection. In modern times, the design is to a larger extent based on scientific approaches such as extreme value statistics on measured water levels and waves. Still, historical observations can be used to complement the relatively short time series of systematic measurements [17–20]. Previously experienced flood events can also inform societies about the consequences of flooding and how societies can be built in a more robust and resilient manner. However, the consequences of a historical storm event cannot be directly applied in a present or future society if the state of the system has changed [21]. The storm events themselves can drive such changes through, e.g., increased risk awareness and implementation of flood protection [22]. System changes may also be driven by processes independent of the storm event, such as societal development, exploitation of coastal areas, and increased property values.

Previous studies have shown that risk awareness and the strength of the collective memory of floods can vary over relatively short geographic distances [23,24], even within regions [10,25], and influence the attitude towards risk reduction measures [25].

Social-science discourses formed the base for explanatory approaches to why local and cultural differences of risk awareness and memory emerge [23,26,27]. First, cultures can be understood as different forms of shared knowledge of actors, which include, e.g., risk perceptions or collective memories of storm floods. Second, a core idea of communicative

constructivism can be applied. There, the development of culture and identity spaces can be consistently deduced to communicative actions [26,28], and cultural knowledge is spread by communication procedures. As communicative actions are bounded locally and socially, communication procedures lead to the development of different forms of locally shared knowledge, i.e., cultures [26,28,29]. Third, the development of new forms of shared knowledge, including cultural change, strongly depends on already existing forms of locally shared knowledge [26].

For the collective memory, stakeholders and communities who collect, archive, remember, and communicate flood histories are important actors [10]. Resources for the collective memory of floods are, e.g., narratives, oral and archived histories, artefacts, material practices in the landscape, media, folk memories, and autobiographical records. The antipodal to remembering is forgetting. Forgetting can be a subconscious decay process, but it may also be an active process [8,9]. Active forgetting can be motivated by trauma or economic interests, in the case of flooding, e.g., house prices or business [10].

This study will first present an integrated description of the physical and societal impacts and consequences of the 1872 storm in Denmark, Germany, and Sweden. This will be followed by an investigation of the response to the storm and its influence on the collective memory and flood risk management policies. For this purpose, we map the organization and knowledge about coastal flood risk management from 1872 until today in the affected countries.

## 2. Materials and Methods

This article is a combination of a synthesis and an original research article about the 1872 storm and coastal management in Denmark, Germany, and Sweden. It is based on an analysis of research articles and grey literature, such as material from archives, private collections, and museums. Most of the literature concerning the 1872 storm is only available in the original languages, Danish, German, and Swedish. This article aims to make this knowledge available for a broader research community and compares the impact and response in the affected countries with a multi-disciplinary approach.

The authors have previously investigated the 1872 storm in their respective countries and disciplines, such as coastal engineering and management, physical geography, environmental history, cultural anthropology, ethnography, and sociology. The main part of the analysis and discussion presented in this article was developed during a workshop held in Lund, Sweden, in October 2019.

## 3. Results

The results of the literature study are presented for three topics: the hydrodynamics of the 1872 storm; the consequences of the 1872 storm, in terms of both damage and recovery; and the role of the 1872 storm in the development of coastal flood risk management. The latter two are presented for each of the countries separately.

### 3.1. Physical Description of the 1872 Storm

On November 13th in 1872, an extreme storm event—often referred to as the 1872 storm—caused severe flooding along the coasts surrounding the western parts of the south Baltic Sea (Figure 2). The peak water levels during the 1872 storm were by far the highest on record, exceeding the observed water levels of the area in the last 100 years by more than a meter [30] (Figure 1).

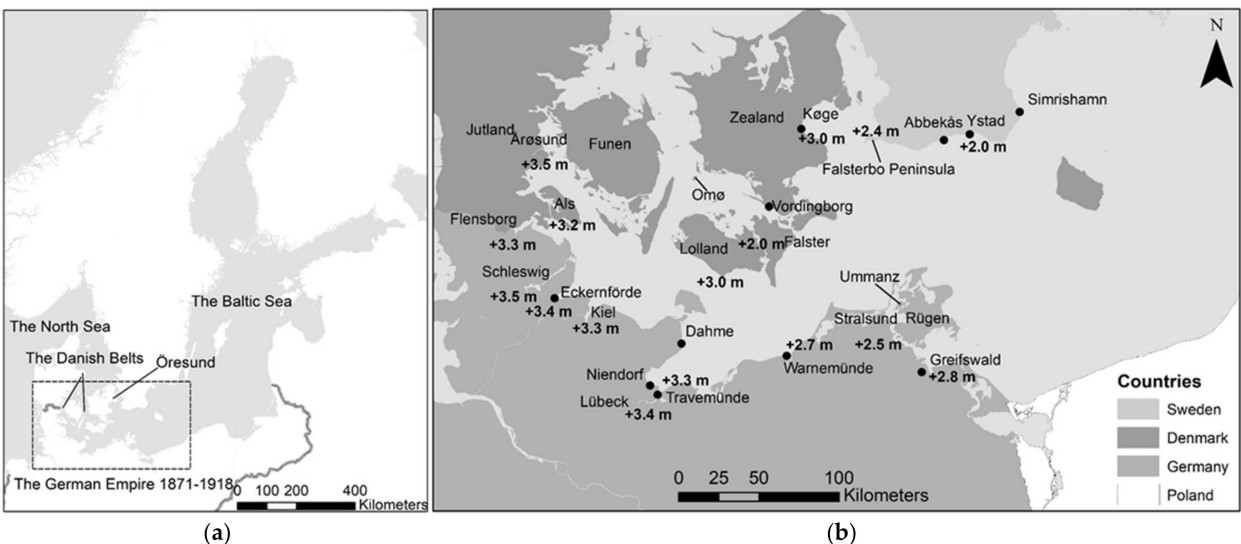

**Figure 2.** (**a**) Overview of the Baltic Sea basin. The land boundaries of the German Empire as in 1872 are indicated. (**b**) Overview of the affected area during the 1872 storm and the measured or reconstructed peak water levels along the coasts.

More specifically, during the 1872 storm, the still water level in Germany reached up to 3.3 m above normal sea level at that time in Travemünde, Kiel, and Flensborg, 3.4 m in Lübeck and Eckernförde, 3.5 m in Schleswig, 2.8 m in Greifswald, 2.7 m in Warnemünde, and 2.5 m in Stralsund [15,31]. In Denmark, the highest observations were made in Årøsund at the southeast coast of Jutland, at 3.5 m above normal sea level at that time, and in Als, at 3.2 m [31,32]. In Køge, on the east coast of Zealand, and Kramnitze Grab, Lolland, the peak water level reached 3.0 m above normal sea level at that time, and in Nykøbing, Falster, 2.0 m [32]. The peak water level at Falsterbo Peninsula in Sweden has been estimated to 2.4 m above normal sea level at that time, based on a flood mark and eyewitness stories [33]. From Ystad harbour on the Swedish south coast, there is an observation of 2.0 m above normal sea level at that time [34].

The Baltic Sea is a nearly closed basin connected to the North Sea by the Danish belts and the narrow sound, Öresund, between Sweden and Denmark. The astronomic tide in the Baltic Sea is negligible (<10 cm), and in the area that was affected by the 1872 storm, coastal flooding is rare [13]. The extreme water levels during the 1872 storm were caused by a sequence of pressure systems over Scandinavia and central Europe that created an unusual wind climate [32].

Rosenhagen and Bork [35] reconstructed the weather from 1 to 13 November based on pressure observations. They found that prior to 10 November, a low-pressure system over the North Sea and Scandinavia generated strong westerly winds that pushed large volumes of water into the Baltic Sea. At this time, the water levels in the southwest Baltic Sea were low, whereas the water levels were elevated along the northeastern coasts. The low-pressure system then moved away east, and a high-pressure system was established over Scandinavia. On 12 November, a low-pressure system moved in over Central Europe, and the southwesterly winds ceased. After a calm period with weak winds, on November 13th, the high-pressure system over Scandinavia and the low-pressure system over central Europe intensified. The large pressure gradient generated strong northeasterly to easterly winds, reaching hurricane strength in the south Baltic Sea. Water was pushed towards the southwest Baltic Sea, and the pressure gradient over the Baltic Sea basin contributed to a further increase of the water levels in the south. The storm surge reached its peak in the morning on 13 November. In the afternoon, the wind speed decreased with a decreasing pressure gradient between the high-pressure and low-pressure systems.

The peak water levels were reached earlier in the eastern part of the south Baltic Sea, slowly moving west [32]. They coincided with large waves along the east coast

of south Sweden, the east coast of the wave-exposed Danish islands, and the German coast [12,32,36,37]. The main contribution to the extreme water levels were the remote winds and the seiches in the Baltic Sea basin, pushing the water towards the southwest [36]. In bays, local wind setup further increased the water levels by more than half a meter [36].

In the gauge records (Figure 1), the 1872 storm stands out as a singularity, and there are large differences in the estimates of the return period. Estimates of the return period on the German Baltic coast range from 180–200 years [38], 1000–2500 years [15], up to 3400–10,000 years [39]. In Sweden, the return period was estimated to 7000 years using a generalized extreme value (GEV) model based on 100 years of water level observations [33]. In Denmark, the return period was far over 1000 years in analyses by the Danish Coastal Authorities [40].

The results of the extreme value analyses depend on the methods used, whether other historical events outside the gauge time series are taken into account or not, and the location along the coast. One issue with these analyses is that measurement records with a maximum extension of about 100–200 years are insufficient to estimate the probability of an event with such a long return period. Further, GEV models assume that the studied block maxima belong to the same distribution (see, e.g., [41]). Since multiple processes interact when the most extreme water levels are generated in the south Baltic Sea, this assumption might be invalid [33].

Instead, studies of other historical storm events can supplement the analysis of the prevalence of storm surges of the same magnitude as the 1872 storm. Several historical storm surge levels have been estimated based on flood marks (Figure 3) and information in German and Danish literature [13,42]. The earliest reliable record of a Baltic storm surge dates back to 1044 [13]. More precise records of storm surges are available from 1304, 1320, 1625, 1694, 1784, and 1835, but they most likely had lower peak water levels than the 1872 storm [15]. In Travemünde, the water level was estimated to have reached 3.1–3.2 m above normal sea level at its peak in 1320 and 2.84 and 2.86 m in 1625 and 1694, respectively [13]. However, it should be noted that these estimates are very uncertain considering the difficulties of estimating the normal sea level at those times. Further, climate change and variability make it difficult to use historical events for extreme value analysis and extrapolation in the future. For instance, the prevailing wind directions in the south Baltic Sea have changed over the last centuries, influencing the probability of extreme storm surges [43,44].

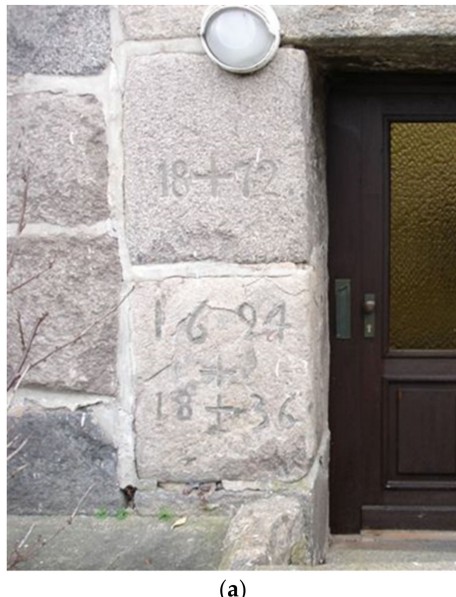

(a)

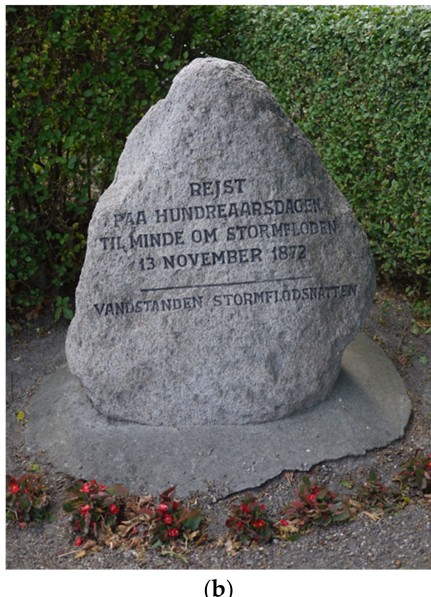

(b)

**Figure 3.** (**a**) Flood marks from 1694, 1836, and 1872 on a building in Schleswig, Germany. Photo: R. Sedlatschek; (**b**) Flood mark at Gedesby church (Gedser, Denmark) to memorize the 100th anniversary of the 1872 storm surge. Photo: Wikipedia.org.

Since 1872, the mean sea level in the Baltic Sea has increased. During the 20th century, the average increase of the mean sea level in the southwest Baltic Sea was $1.2 \pm 0.1$ mm/yr [16]. In the last decades, the sea level rise has accelerated; in 1993–2014, the estimated sea-level change rose to $3.4 \pm 0.7$ mm/yr in the Baltic Sea [45]. On the Swedish and Danish coasts in the southwest Baltic Sea, the sea-level rise is partly compensated by a small post-glacial uplift < 1 mm/yr, but on the German coast, the post-glacial uplift is close to zero [46]. On average, the relative sea-level rise in the study area has been in the order of 10–30 cm since the 1872 storm. The mean sea level is expected to keep increasing at higher rates, implying that extreme water levels in the same order of magnitude as in 1872 will be more common in the future [47].

### 3.2. Consequences of the 1872 Storm

The 1872 storm and the associated coastal flooding are unique events along the coasts of the southwestern Baltic Sea. In this area, where coastal flooding is rare, low-lying areas had been developed without sufficient flood protection before the storm. People both at land and at sea were taken by surprise by the storm. In 1872, there were still no operational storm warning systems that managed to predict the storm and warn the public [48,49]. In Denmark and Germany, 271 people were deceased: 99 on land in Denmark, 63 on land in Germany, and 109 at sea [11]. In Schleswig-Holstein, 15,000 people lost their homes, and 2800 buildings were damaged [11]. In Sweden, at least 23 people lost their lives in the storm, of which 5 were on land and 18 at sea, and more than 100 houses were destroyed [12]. In addition to the devastation on land, 654 ships were damaged in the Baltic Sea and the North Sea during the storm, of which 293 were in Denmark, 122 in Germany, and 56 in Sweden [11]. It is uncertain if the 18 deceased at sea on the Swedish coasts were also accounted for in the numbers of those deceased at sea from Kieksee [11].

In the following sections, the consequences and responses to the 1872 storm are presented for each country separately.

### 3.2.1. Germany

The effects of the 1872 storm in the two German states on the Baltic Sea coast, Schleswig-Holstein and Mecklenburg-Western Pomerania, are relatively well-documented, partly due to the archival work of Heinz Kieksee in memory of the 100th anniversary of the storm [11]. He found that of the total 63 people that died in Germany, 32 were in Mecklenburg-Western Pomerania and 31 in Schleswig-Holstein. Thousands of houses were destroyed, and major damages occurred to coastal infrastructure such as gauge stations along the entire German coast [11]. Of the 122 ships that went down or were damaged along the German coast, 65 were in Schleswig-Holstein and 57 in Mecklenburg-Western Pomerania [11].

In Dahme and Eckernförde on the German coast, the 1872 storm caused large devastation. In the city of Eckernförde, 78 houses were destroyed and 138 damaged, leaving 112 families homeless, but luckily everybody survived [11]. In the small village of Dahme, ten people died in the floods, more than anywhere else in Schleswig-Holstein. According to the local cultural society [50], only 20 houses out of the original 80 or 90 in Dahme were still inhabitable after the flood. More than 50 families with about 300 people became homeless. Most of the livestock, e.g., 350 cows, drowned. Four years later, in 1876, the annual municipal account was titled: "The year IV after the big flood".

As a response to the disaster, only a few weeks after the storm, on 30 November 1872, the "German help association for the needy on the Baltic Sea coast" was founded in Berlin. An appeal went through daily newspapers throughout Germany. The foundation was under the Protectorate of the German Crown Prince, and the call was signed by 173 well-known public figures [11]. Other campaigns, e.g., from the German women's association or local town committees, were also issued in due course of November calling for solidarity and financial support for the Baltic Sea communities.

Since the late 18th century, art and literature had awakened awareness of Germany's Baltic Sea landscape. The Greifswald-born painter Caspar David Friedrich (1774–1840) and the pastor and Greifswald professor Ludwig Gotthard Kosegarten (1758–1818) had founded a new romantic idea of the Baltic Sea [51]. The Romanesque longing came together with the recommendations of physicians. A stay at the coast was regarded as useful for city dwellers' health and recreation because of the clean, fresh air. Accordingly, a large number of seaside resorts were established along the German coasts since the late 18th century. The storm of 1872 not only damaged these resorts but notably also spurred the development of additional ones.

For the small fishing village of Niendorf, the 1872 storm became the turning point in its development into a seaside resort. In Niendorf, 9 out of 24 houses were destroyed, causing four casualties. Streets, gardens, and fields were covered with sand and stones up to one meter [52]. The newspaper reports and appeals for donations made the hard-hit fishing village Niendorf famous throughout the newly established German Kingdom. With the help of donations, new houses and hotels were built to meet the needs of the summer guests soon coming from Lübeck and Hamburg [52]. Already a few years after the storm, in the mid-1870s, seaside tourism flourished in Niendorf. The willingness to donate for the flood victims throughout the empire made the reconstruction possible within a short time.

In Ummanz, a small village 300 km east from Niendorf, the community promoted the damming of the islet, and in 1874, a "ring wall" was built, surrounding settlements and scattered farmlands. The village chronicle of Ummanz reports: "After the devastating storm surge of 1872, the state decided to give Lieschow protection against the destructive floods. A rescuing dike was built with a state subsidy of 30,000 Gold mark". Still today, the community of Ummanz favours the preservation of the existing structure of the old "ring dike", which now is in need of maintenance. Meanwhile, the authorities call for a so-called "cross-bar solution", which would grant protection to the villagers in the centre while leaving their land unprotected [24].

The 1872 storm has been thoroughly described in German literature, inspiring scientists, artists, and authors alike [11,38,53]. For example, in Stralsund, the lime plant caught fire, resulting in a sparking island of flames, which was later illustrated in an oil painting (Stralsundische Zeitung, 14 November 1872). Countless flood marks can be found on buildings in German villages and towns along the Baltic Sea (Figure 3). As early as 1824, the Royal Prussian Government in Magdeburg had issued a decree concerning the marking of exceptionally high and low water levels issued by the ministry of trade in Berlin [54]. The decree might have possibly helped develop the distinctive custom to erect flood marks and memory stones throughout Germany.

Due to the storm's unique magnitude and the resulting high impact in terms of fatalities, destroyed houses, infrastructure, and changes in lifestyles, the collective memory, manifested through rich documentations, museums, and flood marks, seems to be relatively strong in Germany today. Still, some studies show that risk awareness among German experts is relatively low [55]. Experts seem to evaluate storm floods for the near future mainly as a problem for the North Sea and less for the Baltic Sea coast [56].

### 3.2.2. Denmark

In Denmark, the settlements around the Baltic Sea had expanded rapidly during the 19th century [48]. With the urbanization came industrialization and new socio-economical patterns that increased the vulnerability in flood-prone areas and aggravated the consequences of the storm [48].

The 1872 storm mainly impacted the islands in the south Baltic Sea and the southeast coast of Jutland; in total, 52 people lost their lives on Falster, 40 on Lolland, and 3 on Als [11]. In the parish Gloslunde on Lolland, 26 people drowned, and almost all inhabitants became homeless. Apart from the 26 deceased, the local newspaper of Nakskov reported the loss of 80 cows, 8 horses, 40 pigs, and 200 sheep; 6 or 7 houses were gone, 50 houses lost their walls, 10 farmhouses were destroyed, and many more were damaged [57]. Kieksee [11]

further reports that on the peninsula of Hummingen, also on Lolland, 11 persons were found on a floating roof and saved, and the village of Vemmingbund in the Flensborg fjord was completely destroyed, whereupon 70 families with 280 people became homeless. There exists other anecdotal information about more deaths and injuries resulting from the flood in diaries, local newspapers of that time, and written eyewitness accounts found in local archives, but those remain unconfirmed [58].

The local historic archives describe the devastating impact on society and how people helped each other cope as best as possible. It is remarked that the storm surge had a very short duration and that the flooded areas decreased rapidly after the storm had peaked. Stories from the small islands south of Funen tell that the reason many people survived was that they lived in timbered houses. When the flood hit the houses, the clay was washed away, but the wooden frame kept standing so that the houses' occupants could stay safe on the attic or roofs. In other places, many houses were completely destroyed, and large agricultural areas were so damaged by saltwater that it took years for them to recover.

The storm received a lot of attention in newspapers, and private donations were collected to support the people affected by the storm, especially on Lolland and Falster. There are more than 50 flood marks in Denmark to remember the event and the victims or to mark the water level during the storm. Most of the flood marks are found in Lolland, Falster, and the south part of Jutland. In 1972 on the 100th anniversary of the 1872 storm, a flood mark was placed at Gedesby church, on Falster, to raise awareness about the catastrophe (Figure 3b). At Gloslunde's rectory, there is a museum about the flooding with a memorial to the 26 people that lost their lives within the parish.

In Denmark, the storm also left morphological traces, which are visible still today. At the sandspit Feddet in Faxe Bay, the impact of the 1872 storm was investigated with ground penetrating radar (GPR) and optically stimulated luminescence (OSL) analyses to map and date the different sand layers of beach ridges on the spit [37]. The survey revealed that the storm had formed a high storm berm and that sediment had been transported 250 m inland at 2.5 m above MSL through overwash processes.

3.2.3. Sweden

There has been little documentation about the effects of the 1872 storm in Sweden until the recent publications by Feldmann Eellend [59] and Fredriksson et al. [12]. The 1872 storm mainly impacted the southern and eastern coast of the county Skåne, where at least 23 people lost their lives and more than 100 houses were destroyed [12]. Among the deceased, five people were pulled out from land by large waves in Simrishamn on the east coast, whereas the others were lost at sea. Figure 4 shows a painting of a shipwreck in Simrishamn, where high water levels coincided with large waves.

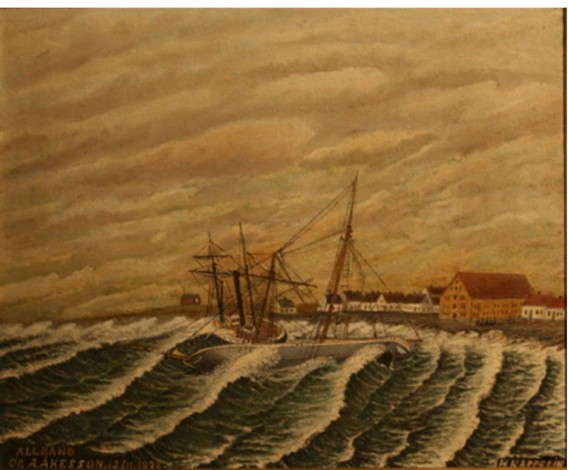

**Figure 4.** Oil painting of the ship Albano being wrecked in Simrishamn, painted by H. Kappelin.

At this time, the coast was mainly inhabited by fishermen and their families. The number of fishermen increased significantly during the second part of the 18th century due to the abolition of the royal fishing monopoly. Fishing constituted a much larger part of the Swedish economy than it does today and increased further in 1840–1890 due to rapid population growth [12]. The fishing villages on the Skåne coast had been growing, and several new villages were established during this period. During the 1872 storm, the largest damage was caused in the fishing villages along the coast, where the fishermen lost their houses, boats, and fishing equipment, and several harbours were destroyed.

After the storm, the harbours were rebuilt with financial support from the government and private donations. On 19 November—only a week after the flooding—a mayor from the Swedish army corps of engineers arrived in Simrishamn and started to plan the harbour's restoration [12]. Existing local harbour and dike associations facilitated voluntary work [60,61]. At the Falsterbo Peninsula, the seaweed dikes destroyed during the storm were quickly repaired [62]. Along the coast, damaged houses were rebuilt with more resistant constructions and high bases to protect the walls against water [62,63].

Although part of the poor coastal population lost their houses, belongings, and income source, the coastal societies recovered quickly through governmental subsidies, voluntary work, and charity [12]. In the fishing village of Abbekås, the storm has been described as the transition from ancient to modern times [63]. The renovation of the harbour was meant a modernization that contributed to economic growth. However, the village's flourishing period was disrupted by a new storm in 1904 that again damaged the harbour [63].

Funds were raised to support victims of the flood both in Sweden and abroad [12]. In the end, there was more money donated than needed to rebuild and repair the damages caused by the storm. The surplus was invested in a fund to help poor fishermen and their families on the southeast coast of Skåne [12]. Still today, grants are distributed yearly from the fund to the local lifeguard association. The fund is an institution to memorize the 1872 storm. But apart from the fund, there is little reminder about the storm in Sweden. Only one flood mark, a stone with an inscription at the Falsterbo Peninsula, is known to the authors. The stone has a rather secluded placement along a street where it is barely noticeable [33].

The storm is mentioned in some local museums and homeland literature [62,63] but was until recently largely unknown to the public. However, recent research and attention in newspapers, radio programs, and art exhibitions have increased awareness about the storm, demonstrating that the collective memory of an event can be revitalized.

### 3.3. Organization of Flood Risk Management from 1872 until Today

The countries affected by the 1872 storm, Germany, Denmark, and Sweden, have different coastal flood risk management organizations. Since 2007, though, they are all covered by the EU Floods Directive (2007/60/EC). The purpose of the EU Floods Directive is to establish a framework for assessing and managing flood risks, aiming to reduce the adverse consequences for human health, the environment, cultural heritage, and economic activity associated with floods. The directive requires the member states to assess if their coastlines are at risk of flooding. The risks should be mapped, and adequate measures should be taken to reduce the flood risk. However, the directive has been implemented from very different starting points, starting points that in Denmark, Germany, and Sweden were partly influenced by the response to the 1872 storm.

In the following sections, the coastal flood risk management in each country is described from 1872 until today.

### 3.3.1. Germany

Two German Federal States, Schleswig-Holstein and Mecklenburg-Western Pomerania, share the Baltic Sea coastline of Germany, where more than 1300 km$^2$ of flood-prone coastal lowlands is home to about 235,000 people with 10 billion € of capital asset value [64,65]. A combination of dikes, dunes, beach nourishments, and other flood defences such as

mobile and sheet-pile walls protect the coastal flood risk areas [64,66]. The responsibilities for the embankments are shared among local water boards, municipalities, and state governments.

In Schleswig-Holstein, coastal protection was already in place before the 1872 storm. In the late 1860s, the Prussian government had initiated a coastal flood defence programme as socio-cultural aid for the newly established province of Schleswig-Holstein [67]. One of the first measures built in 1868–1869 was a 5.5 km long embankment in front of Dahme and the Oldenburger Graben, a valley behind Dahme. The design was probably based on observations from the last extreme storm surge in December 1836, which reached 2 m above normal in Dahme. The embankment was built from sandy material with relatively steep slopes on top of existing dunes and beach ridges. In 1870, a local "Sea Association" was founded to maintain the embankment.

During the 1872 storm, only three years after the Dahme embankment's finalization, water levels in the area rose to about 2.8 m above normal. The embankment was not designed for these hydraulic loads and breached in several locations, among others directly in front of Dahme [11]. Not only the village but the whole coastal lowland Oldenburger Graben was inundated. The embankment from 1869 may have abetted the high number of fatalities in Dahme compared to, e.g., Eckernförde. In Dahme, the water levels rose rapidly after the breaching, whereas, in Eckernförde, they rose more steadily and gave the inhabitants time to react.

There were no warnings issued to the public, as the storm took place just at the beginning of the development of meteorological institutions and weather services. The German Meteorological Society (DMG) was not founded until 1883. However, meteorological services run by the German states had been established previously. The most prominent was the North German Naval Observatory founded in Hamburg in 1868, which was granted the status of imperial authority in 1875 as the "Deutsche Seewarte" (German Naval Observatory); it primarily served maritime meteorology in support of ocean shipping [68]. In addition, it soon developed into a weather information centre that provided weather reports to numerous other meteorological institutions in Germany and beyond.

The 1872 storm can be seen as a turning point for coastal flood defence along the Baltic Sea coast of Germany. Less than one month after the flood, the Prussian government passed a decree with design criteria for a comprehensive coastal flood defence programme in Schleswig-Holstein [67]. One criterion was that new embankments should be erected sufficiently landward of dunes and beach ridges instead of on top of them. It was recognized that sandy shorelines retreat during storm surges, thereby eroding and destabilizing the existing embankments erected at the top of the dunes and beach ridges. After evaluation of the 1872 hydraulic loads, it was further stipulated that:

- the height of the embankment should be about 5.0 m above mean sea level,
- the crest width should be about 3 to 4 m,
- the outer slope should have a gradient of 1:6, the inner slope 1:2, and
- that the embankment should have a cover of at least 0.6 m of erosion-resistant material such as clay.

On 24 April 1873, the "Law, concerning the granting of funds for the removal of the flood damage caused by the storm surge of the Baltic Sea on 12 and 13 November 1872 and the execution of dikes and bank protection works on the coasts of the provinces of Pomerania and Schleswig-Holstein" prepared by the Prussian government came into force. The law made 2.5 million thalers available for individuals and communities. In individual cases, up to 250,000 thalers could be granted without any obligation to pay back, and there were also loans with interest and amortization [11].

In all, about 70 km of embankments were erected in Schleswig-Holstein until 1882 [67]. These defences protected about 145 km$^2$ of lowlands. Local associations were founded with the task to maintain the embankments and secure the drainage of the lowlands. In these associations, all affected landowners were members with the power of co-decision according to their landholding. The Prussian design criteria issued in 1872 closely reflected

the present design of state embankments. However, the design criteria were not fully implemented, mainly due to financial constraints. Most of the new embankments were erected behind natural dunes and beach ridges, but the embankments' mean height was mostly about 4 m above mean sea level; crest widths were among 2–4 m, and the outer slopes had gradients between 1:3 and 1:6. The inner slopes were normally steeper than 1:2.

As documented in the two cases of Niendorf and Ummanz, flood protection was not implemented in all flooded areas [24,25]. For example, in Niendorf and the nearby spa Timmendorfer Strand, the storm's consequences were gradually forgotten. With growing prosperity and continued investment in infrastructure, hotels and property were developed with open verandas facing the shoreline largely without defence measures.

In 1972, the Schleswig-Holstein State Government took over the technical and financial responsibility for a large part of the coastal flood defences from the local associations. In 1977, with the second update of the master plan for coastal flood defence and coastal protection, most state embankments along the Baltic Sea coast were deemed unsafe according to the safety standard [69]. The safety standard was defined as the 1872 storm surge water level plus 0.5 m to consider sea-level rise from 1872 to about 2075. The same safety standard for state embankments was introduced in Mecklenburg-Western Pomerania in 1995 [66].

The deterministic safety standard based on the 1872 storm remained valid until 2012. With the fourth update of the master plan in 2012 [64], Schleswig-Holstein adopted one uniform, statistically derived safety standard for the North Sea and the Baltic Sea coastlines as well as for the Elbe estuary. According to this standard, state embankments should withstand a storm surge water level with a yearly probability of 0.005 (return period 200 years).

The main reason for replacing the deterministic approach with a statistical one was the implementation of the EU Floods Directive. The EU Floods Directive from 2007 forced a harmonization of techniques for the estimation of design criteria. Schleswig-Holstein and Mecklenburg-Western Pomerania have established a joint method based on the probability of extreme storm surge events. The aim is to create a uniform, cross-national basis for the design, dimensioning, and safety assessment of coastal defence measures. According to this directive, the assessment and management of flood risks should be conducted based on a combination of the probability of a flood event and its potential adverse consequences.

The shift from a deterministic to a statistical approach led to a problem on how to consider the 1872 storm. The 1872 storm surge represents, at least in Schleswig-Holstein, a singular event, classifying it as an outlier from a statistical point of view [15]. In the extreme value analysis of yearly highest water levels covering the period 1826–2016 from the Travemünde gauge station, the 1872 storm event was excluded [70]. The resulting design storm surge level was 0.8 m lower than the 1872 storm peak water level [70].

In Schleswig-Holstein, the State Development Plan is currently being updated. It will provide areas of preference for coastal flood defence and climate change adaptation along the coast. The areas of preference for coastal flood defence comprise all coastal lowlands that are not adequately protected against flooding due to storm surges. Here, the erection of new physical structures is generally prohibited. The areas of preference for climate change adaptation along coasts comprise 150 m broad zones behind cliffs, dunes, and beach ridges as buffer zones for sea-level-rise-induced coastal retreat and 50 m broad zones behind state embankments as room for future strengthening campaigns.

### 3.3.2. Denmark

In Denmark, there was some flood protection in place before the 1872 storm that the landowners had constructed with the primary purpose to reclaim land and protect agricultural areas. However, instead of decreasing the flood risk, these dikes contributed to an increased flood risk since people had relocated to more low-lying areas without sufficient flood protection. Many of the dikes built during the 19th century were built by people with limited knowledge about dike construction. The dike constructions were often weak, they consisted mostly of sand, and often, water was able to flow in from the

surrounding areas. In some places, the dikes were constructed with too steep slopes and without sufficient vegetation cover [48]. Most dikes were only between 1.5 to 2 m above the mean sea level and were overtopped during the 1872 storm.

The Danish Meteorological institute was just opened in April 1872 but did not manage to predict the flood. However, the flood highlighted the importance of meteorological science and warning systems, thereby supporting the new institute's work. Regular weather forecasts became standard from the late 1870s and forward. The 1872 flood might be seen as having influenced the development of modern warning systems, meteorological science, and the way that the Danish Meteorological Institute communicates to the public [48].

After the storm, there was an immediate response to improve flood safety. Existing dikes were reinforced, and two new dikes were constructed on Lolland and Falster with financial support from the government. In May 1873, a new law was passed authorizing the building of a 63 km long dike on Lolland and a 17 km long dike on Falster, which in this area are considered to be very long [57]. The law (No. 69 of 23/05/1873) committed the national government to finance 2/5 of the dike on Falster and 1/4 of the dike on Lolland. The rest of the costs were covered by the regional government and property owners protected by the dike [71]. The dikes were erected by 150 soldiers from the Danish Army's Corps of Engineers.

The economic support from the national and regional governments and the short legislation process were effects of the great attention the flood had received by the public at that time. However, it still took several years before the two dikes were finished. The dike on Falster was finished in 1875, and the one on Lolland in 1878. The storm led to an increased awareness of the importance of dikes, especially how they should be built. The two new dikes on Lolland and Falster were built according to modern engineering knowledge. Compared to the previous dikes, they were built with milder slopes and stabilizing vegetation. For both of the dikes, dike boards were established to ensure that the dikes' safety levels were maintained.

Along the Danish coast, other minor dikes were built to protect against a similar storm. However, as time passed without any storm surges reaching close to the 1872 storm surge level, flood risk awareness diminished. For example, the dike on the island of Omø that was constructed after the 1872 storm surge was later partly removed (Figure 5).

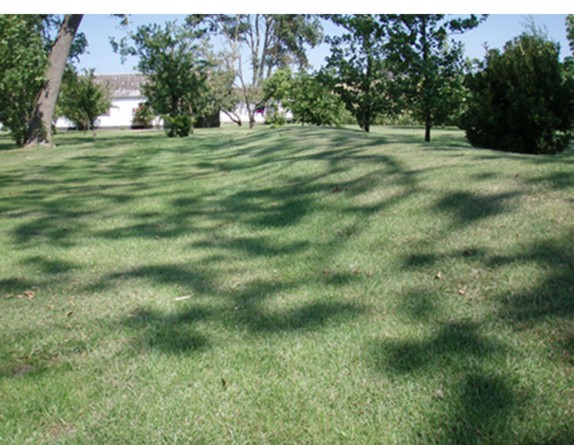

**Figure 5.** Remnants of a dike on the Danish island of Omø constructed after the 1872 storm, but later partly removed. Photo: Per Sørensen.

Also, in other areas influenced by the 1872 storm surge, landowners and planning authorities either have forgotten the 1872 storm or now accept a significantly higher risk of being flooded. Many of these flood-prone areas were previously used as farmland and for storage houses in harbours. Growing wealth from the 1950s onwards started a slow descent of the coastal towns from the higher ground towards the sea. Recreational summer cottages were built on low-lying farmland, most of them unprotected from flooding. In the last

20 years, some of these areas have transformed into permanent residential areas, inhabited all year round.

As before the 1872 flood, coastal flood protection is still a private responsibility in Denmark. New flood protection should be planned, funded, and later maintained by the home- or landowners they protect. However, local governments can contribute with funding if they wish. From the last ten years, there have been several examples where municipalities have supported the building of new dikes through helping with planning and advice free of charge. However, the construction costs have been paid by the landowners.

This tradition of private responsibility for coastal flood protection likely grew out of the fact that most dikes in the 19th century were built to gain more farmland, not for flood protection [48]. The government's economic support to the dikes at Lolland and Falster after the 1872 flood was then and is still today an unusual action. Therefore, the response after the 1872 flood does not seem to have influenced the laws related to flood protection.

In 1991, the Danish Storm Surges Council (later renamed The Danish Storm Council) was formed. It was formed as a response to the increased number of damages from floods during the 1980s. Standard Danish house insurance did not cover those damages, and therefore, a natural disaster fund was suggested [72]. Today, the storm council can grant compensation for economic losses due to flooding from a severe storm surge, defined as having a return period of 20 years or more. The fund is financed through the house owner's fire insurance.

Nonetheless, today the 1872 storm is used as a baseline for flood risk assessment in the affected areas. In both the first and second cycle of the work related to the EU Floods Directive, the extent of the 1872 flooding determined the areas included. The calculated risk builds on the historical water level in 1872. Thus, the 1872 storm has directly influenced which areas are appointed as high-risk areas and should develop risk management plans [73].

There is now a growing debate about the fairness of continuing having flood protection as a private responsibility in Denmark. The governments of larger cities plan and fund larger flood protection projects in response to climate change. This is not possible in smaller and less populated municipalities, creating inequality between different parts of the country. Furthermore, private planning of flood protection reduces the coordination between individual projects. These arguments have been put forward in media with the argument of changing the law. In March 2020, the Minister of the Environment stated a need for a national strategy for climate adaptation, focusing on coordination of flood protection, which indicates a growing awareness on the national level of a need for better coordination and support of flood protection.

Baron [74] studied the local attitude towards flood protection on the Danish islands. Still today, the 1872 storm stands very strong in the inhabitants' collective memory on Lolland and Falster. In interviews with people from those areas, they all mention the 1872 storm. Meanwhile, inhabitants of Vordingborg on the south coast of Zealand rarely mention the event, although they also live in flood risk areas that were affected by the 1872 storm. Baron [74] found that when people on Lolland are told they live in a high-risk area, they accept it and often refer back to the 1872 flood themselves. But when people in the appointed risk areas in Vordingborg are presented with the same information, they find it unrealistic.

In 2006, Vordingborg was hit by a storm surge, which flooded several houses. As a response, two new dikes were built. In the planning of those dikes, the coastal engineers included the 1872 storm in their models. However, when the projects were presented for the local homeowners, the design based on the 1872 storm was turned down. The homeowners found the crest elevation too high and decided to reduce the proposed height to reduce cost and keep their sea view. The finished dikes can withstand a flood like the one in 2006, but not like the 1872 storm. This means that the areas behind the dikes are still appointed as high-risk areas according to risk assessments following the EU Floods Directive.

However, on Lolland and Falster, differences in memories of the 1872 flood also create disagreements about the present flood protection level. An example is an ongoing conflict about cycling on the dike on Falster. Some argue that it weakens the dike, while others do not see it as a problem as they cannot imagine the water ever rising to the top of the dike. Those against cycling refer back to the 1872 flood. Meanwhile, those in favour of cycling do not mention this historical event. This debate demonstrates that the 1872 storm is, to many, a historical event, not something that forms their risk perception today.

### 3.3.3. Sweden

At the time of the 1872 storm, no national or regional organization in Sweden was responsible for coastal flood protection [12]. Neither were there any predictions or warnings about the storm. In 1872, the decision had been made to establish a national centre for meteorological studies in Sweden, but it was not until 1905 that a storm warning system was operational [49].

Despite the lack of national or regional flood risk management, historical documents indicate that the local population of the low-lying Falsterbo Peninsula in southwest Skåne was aware of the flood risk [59]. The two cities of Skanör and Falsterbo were located at the peninsula's highest elevations and surrounded by dikes serving both as cattle fences and flood protection [61]. The dikes were constructed of seaweed with crest heights of 1.65 m above the ground and maintained by local dike associations [62]. During the 1872 storm, the dikes breached, and several houses were damaged when the water dissolved the clay material in the walls. However, the damages were limited, since most houses were located at the most elevated parts of the peninsula. Since 1872, thousands of houses have been built in flood-prone areas without flood protection [12]. The seaweed dikes at the Falsterbo Peninsula have not been maintained since the early 20th century; instead, they are preserved and protected as fragmented ancient remains.

There is no indication that the 1872 storm has influenced the organization of coastal management in Sweden. There was an immediate response to the storm at the local level, reinforcing the seaweed dikes and houses, but no policy changes. With time, risk awareness in the coastal societies—which have been subject to extensive migration of people without a connection to the local environment—has decreased [30].

For many years, low-lying coastal areas were developed without considering the flood risk. Still today, no organization is responsible for implementing coastal protection [75]. The municipalities in Sweden have a planning monopoly and can therefore decide on the land use within the municipality's borders. However, they do not have any legal responsibility to protect their citizens against coastal flooding. Landowners should, in case it is desired, organize coastal protection themselves.

The Planning and Building Act regulates spatial planning by the municipalities. However, it was not until 2008 that a condition was added to the act stating that the land should be appropriate to develop concerning erosion and flooding (2nd chapter, 5th paragraph). In 2018, there was another addition to the act regarding flood and erosion risk in comprehensive planning (3rd chapter, 5th paragraph). It was stated that municipalities should give their view on climate change-related risks in developed areas concerning flooding and erosion and how they can be mitigated or eliminated. However, municipalities still do not have any legal obligation to protect their citizens against coastal flooding.

On the other hand, with the implementation of the EU Floods Directive, there have been changes to the legislation that lift at least the responsibility of flood risk management plans to higher organizational levels. The Swedish Civil Contingencies Agency carries out the implementation of the directive. However, in the first cycle of 2009–2015, coastal flood risk was, against the directive, left out from the national analysis of flood-prone areas [76]. In the second cycle, 2016–2021, coastal flood risk is included, and 16 areas at risk of coastal flooding have been identified [77]. In 2020–2021, the County Administrative Boards will make risk management plans for the identified risk areas. However, the

County Administrative Boards do not have any funding or legal rights to implement flood protection in the municipalities.

Up until now, coastal flood and erosion risks have typically been dealt with on a landowner or municipal level. Some municipalities in south Sweden have decided to implement coastal protection to protect their infrastructure, larger developed areas, or maintain beaches, although they do not have a legal obligation to do so. In 2020, e.g., Vellinge municipality at Falsterbo Peninsula was granted a permit from the Environmental Court to build a dike ring to protect the peninsula's urban areas. The 1872 storm was then used in the risk analysis but not as a design criterion for the coastal defence [78]. When private landowners implement coastal protection, it is often done on a small scale for one or a few houses without a proper design procedure or legal permits.

Coastal protection measures require a permit from the Environmental Court according to chapter 11 in the Swedish Environmental Code. If the measure has a smaller spatial extent than 3000 m$^2$, an application to the County Administrative Board is sufficient (unless a permit from the Environmental Court is required considering the impact on the environment and stakeholders). The design criteria are typically proposed by the applicant and assessed by the Environmental Court or the County Administrative Board, from case to case.

Municipalities' and national agencies' work related to climate change adaptation, the implementation of the EU Floods Directive, and recent research about the 1872 storm have gradually drawn attention to coastal flood risk in Sweden. Following the EU Floods Directive, the 1872 storm has been considered in the flood risk analysis in two out of four flood-prone locations within the affected area [34,79–81].

## 4. Discussion

The response in terms of flood risk management to the 1872 storm event was different in the affected countries. In Denmark and Germany, dikes were erected to protect the flooded areas from similar events [11,71]. However, the storm was more or less forgotten in Sweden, and thousands of houses were built in flood-prone areas without flood protection [12]. In a comparative study of coastal communities in Germany and Portugal, Martinez et al. [25] found that the culture and related socio-economic and political circumstances of a community significantly influenced and shaped coastal protection measures. In the following, the 1872 storm and its influence on coastal management in Denmark, Germany, and Sweden are discussed in terms of collective memory, as well as from economic and political circumstances in the affected areas.

### 4.1. The Collective Memory of the Storm

There is large variability in the collective memory of the 1872 storm in the affected countries. In Sweden, there is almost no memory of the storm; in Denmark, there is locally a strong collective memory of the storm; whereas, in Germany, there is a relatively strong memory along the entire Baltic Sea coastline, manifested through rich documentation, flood marks, research studies, and flood defence design guidelines. We have identified several factors that might have influenced the development of the different memory cultures: the level of background knowledge about coastal flooding at the time of the flooding; the organization of coastal and flood risk management at the time of the flooding; the presence of flood marks and dikes today; and the extent of the disaster.

In Denmark and Germany, background knowledge about coastal flooding was stronger at a national level compared to Sweden, based on their flood experiences from their North Sea coasts. However, the inhabitants of the Falsterbo Peninsula in Sweden also had background knowledge about coastal flooding [59]. The damages at Falsterbo Peninsula were small compared to many other villages along the coast [12]; still, the Falsterbo Peninsula is the area with the most documentation about the storm and the only location with a memory mark. To our knowledge, the Falsterbo Peninsula was also the only place in Sweden where dikes were (re)built after the 1872 storm. The results indicate that societies with

more socially shared background knowledge about floods had a greater risk competence manifested in the implementation of coastal flood protection after the storm and in the documentation about the storm.

In Sweden and Denmark, flood risk management was decentralized at the time of the 1872 storm. There were no organizations in place responsible for flood risk management or documentation of the 1872 storm. In Germany, on the other hand, the transfer of coastal protection duties from private initiatives to authorities may have contributed to the relatively strong memory of the storm. This organizational change was coupled with the foundation of the German Empire in 1871. Otto von Bismarck, the first chancellor of the German Kingdom, had had his first public office as dike warden during the 1845 major flood at the river Elbe. However, more investigations are needed regarding Bismarck's potential role as a symbol of the flood risk management culture in Saxony and how this influenced the development of government-led defence programs for Germany's coastlines.

The memory of the 1872 storm is manifested in several forms along the south Baltic Sea coasts, e.g., paintings, drawings, photography, museums, flood marks, and literature. In the years after the storm, there were also examples of flood marks in the landscape, such as overwash fans and erosion scarps [37,82]. However, the landscape memories of storms are often erased when sediment transport processes slowly restore lost beaches and dunes. On the contrary, flood marks and dikes are persistent memories in the landscape that remind about the storm to this day. The absence of dikes and flood marks in Sweden may have contributed to the collective forgetting of the storm.

The collective memory has probably also been influenced by the extent of the damage during the 1872 storm; e.g., the memory is stronger in Germany and Denmark compared to Sweden, and on Lolland compared to Vordingborg locally in Denmark.

### 4.2. Economic and Political Aspects Influencing the Response to the 1872 Storm

The organizational differences among the affected countries influenced not only the memory of the storm but also the response to the storm. Germany was the only country with a centralised (Prussian government) response after the calamity. There, design guidelines and funding solutions were quickly put in place. In Denmark and Sweden, the national government still provided some support, e.g., construction of two dikes on Lolland and Falster in Denmark and rebuilding the harbour in Simrishamn, Sweden. Other actions, such as restoring the dikes at Falsterbo Peninsula, Sweden, and supporting and reconstructing smaller harbours, were mainly organized and funded by private initiatives.

The differences between a central and decentralized organization of coastal management have also manifested in the maintenance and reinforcement of the coastal protection. In Germany, coastal protection has regularly been reassessed, strengthened, and extended, whereas the maintenance of coastal protection was interrupted in Sweden. There, local flood risk awareness was lost with societal changes and migration to the coastal communities.

In Denmark, the economic dependency on flood-prone areas is reflected in the local attitude towards coastal protection. Lolland and Falster, which were the most damaged areas of the 1872 flood, gained and still gain their main income from agriculture on the low-lying, flood-prone areas behind the dikes. In addition, income from tourism, mainly along the coast, has developed since the 1872 storm. On Lolland, there is more focus on keeping the dikes in good repair than in most other areas in Denmark, which might be partly explained by the collective memory of the storm and partly by the economic values at stake. Furthermore, the areas affected by the 1872 storm in Denmark are characterized by being localized far from the major cities. These areas that struggle with social problems and diminishing populations are currently not in the focus of national politics. This might explain why the local population actively takes on the responsibility of dike maintenance through the local dike associations. They feel that they need to be able to handle future floods themselves and do not expect to be prioritized by national emergency management, which they think mainly focuses on the coastal areas in and around Copenhagen [74]. The

national focus on floods increased in 2011 due to an extreme cloud burst in Copenhagen and later in 2013 due to a storm surge hitting a large part of the coast of northern Zealand.

A similar pattern can be seen in Germany. Along the German Baltic Sea coast, the agricultural population is aware of the danger of storm surges, and the memory of the 1872 storm calamity is still alive. Consequently, the coastal lowlands used for agriculture are protected by dikes, either from the state or from local water boards. Today, in Germany, coastal flood risk management is typically seen as a public task (in the coastal states' responsibility) with a strong focus on the North Sea. As such, coastal municipalities and water boards along the Baltic Sea coast often feel neglected or left alone with the challenge. In the German seaside resorts and the port cities, respective economic interests often compete with coastal flood defence interests. Here, perception and awareness for coastal flood hazards and the memory of the 1872 surge are much less pronounced—possibly suppressed–and it is more challenging for the authorities to implement appropriate flood risk management.

*4.3. Influence of the 1872 Storm on Flood Risk and Coastal Protection Today*

Since 1872, there has been extensive development in several of the flooded areas. In some areas, the development has been accompanied by flood protection measures; elsewhere, flood-prone areas have been developed without flood protection or with designs that would not withstand the 1872 storm.

In Schleswig-Holstein, the strict design criteria issued less than a month after the 1872 storm were soon relaxed due to economic factors [67]. Already with the first measures in 1874, financial constraints resulted in deviations from the adopted design criteria. Only ten years after the 1872 storm, the public construction officer acknowledged that the newly erected embankments would probably not withstand an event of the same magnitude [83]. About half a century later, the high-risk areas in Dahme in front of the 1876 embankment were overbuilt [84]. When the strict design criteria were issued only a month after the storm, the necessity, or priority, of using private and public means for recovery and risk management was widely accepted. But the memory and motivation faded as people moved on, and the flood defences had to compete against other interests.

Recent research shows that the collective memory of the 1872 storm still influences the attitude towards flood protection measures. When comparing Ummanz and Niendorf in Germany [24] and Vordingborg and Lolland in Denmark [74], the collective memory of the storm is manifested in greater risk awareness and willingness to accept the negative effects of flood protection measures. The results indicate that strong collective memory of the 1872 storm correlates with more risk-aware societies. The collective forgetting of disasters is a threat against robust risk assessment and sustainable urban planning. However, the example from Sweden shows that the collective memory of the 1872 storm could be recreated through research and communication. The recent attention in media and art exhibitions has brought the 1872 storm into a modern narrative that will likely strengthen flood risk awareness.

Still, more research is needed to explore the probability of the 1872 storm and how the probability and resulting flood consequences are influenced by climate change and climate variability, also taking socio-economic developments into account. Therefore, not only the peak water levels but also the temporal behaviour (i.e., the hydrographs) of extreme coastal events such as the 1872 storm are required [85,86].

Since 1872, the two components of risk—probability and consequence—have both increased due to sea-level rise and coastal development without sufficient flood protection. If the 1872 storm were to repeat itself today, there would be extensive economic damage. Still, there have been a couple of changes in society that might limit the number of casualties. Meteorological prognosis and warning systems are in place that can predict an event like the 1872 storm several days ahead. Warnings would allow time for evacuation and information about how to act when the flood strikes. If modern houses were flooded, they would be more stable and better withstand the waves and water flows, which were crucial factors

for surviving the 1872 storm. Furthermore, higher insurance coverage would mitigate the economic effects on the local population.

## 5. Conclusions and Recommendations

The objectives of this study were to describe the impact and consequences of the 1872 storm in Denmark, Germany, and Sweden and how they have influenced coastal flood risk management. The literature study results confirm that the 1872 storm was an extreme and unique event in the south Baltic Sea. The extent of the damage was unprecedented, and there have not been any coastal flood events of the same magnitude after it. The consequences of the storm in Denmark and Germany were more severe than in Sweden, with significantly larger destruction and numbers of casualties. In Denmark and Germany, the 1872 storm has also been remembered to a larger extent and influenced risk awareness locally and regionally.

In summary, there are large differences in flood risk management in the affected areas, both regionally and locally, that can be traced back to the 1872 storm. In general, the level of protection and implementation of coastal defence systems are largest in Germany, followed by Denmark, and are almost absent in Sweden. However, it is a chicken-or-egg question to conclude whether the collective memory has influenced the implementation of flood protection or the other way around. The collective memory of the 1872 storm correlates with the background knowledge at the time of the flooding, damage extent, and response to the storm. In areas where flood protection was installed after the storm, the collective memory is stronger, and risk awareness and acceptance for flood protection measures are greater.

The centralized organisation of coastal protection at a state level in Germany has contributed to consistent management and reinforcement of the coastal protection installed in response to the 1872 storm. In Sweden and Denmark, the decentralized organisation depending on dike associations resulted in local variability of dike maintenance connected to the collective memory of the storm and the economic dependency on the flood-prone areas. In general, flood risk awareness and acceptance of coastal protection measures are greater in agricultural areas compared to areas dominated by the tourism industry.

However, it seems that the global challenges associated with climate change adaptation along coasts and the implementation of the EU Floods Directive are slowly evening out these differences. These processes appear to have revitalized the memory of the storm, as well as the discussions about the role of the 1872 storm in flood protection design.

Based on this study, the following recommendations are made with the purpose to raise risk awareness in flood-prone areas and promote sustainable and reliable coastal management:

- Include historical storms in discussions and communication about flood risk and coastal management with the public in the affected areas. The consequences of the storms can be illustrated both narratively, by narratives from that time, and visually, as how the water levels during the storm would affect today's society.
- Make memory marks, museums, and exhibitions that keep the collective memory alive and more visible. Existing documentation can be highlighted, and new documentation can be created to revitalize the memory of disasters.
- Further investigate how information on historical storm surges, and especially the exceptional storm surge of 1872, can be used to develop appropriate and sustainable design criteria in the future.

**Author Contributions:** Conceptualization, C.H., G.M., and J.J.; writing—original draft preparation, C.H., J.L.A.H., G.M., J.J., N.B., T.H., A.K., A.A., B.A., P.S. and M.L.; writing—review and editing, C.H., J.L.A.H., G.M., J.J., N.B., T.H., A.K., A.A., B.A., P.S. and M.L.; project administration, M.L.; funding acquisition, M.L. and C.H. All authors have read and agreed to the published version of the manuscript.

**Funding:** Caroline Hallin and Magnus Larson acknowledge financial support from FORMAS (project ID 2018-00288).

**Institutional Review Board Statement:** Not applicable.

**Informed Consent Statement:** Not applicable.

**Data Availability Statement:** The data presented in this study are available on request from the corresponding author.

**Acknowledgments:** Not applicable.

**Conflicts of Interest:** The authors declare no conflict of interest. The funders had no role in the design of the study; in the collection, analyses, or interpretation of data; in the writing of the manuscript; or in the decision to publish the results.

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
