# Peer review of "A Comparative Study of the Effects of the 1872 Storm and Coastal Flood Risk Management in Denmark, Germany, and Sweden"

_water, doi:10.3390/w13121697_

Round 1

Reviewer 1 Report

The paper is well written and interesting to read.

First, I believe there is some overlap between this paper and the following (common co-author): Grit Martinez, Susana Costas, Óscar Ferreira (2020). The role of culture for coastal disaster risk reduction measures: Empirical evidence from northern and southern Europe. Advances in Climate Change Research, Volume 11, Issue 4, 297-309, https://doi.org/10.1016/j.accre.2020.11.001.  I find it odd that this paper is not referenced in the reference list.  Is there sufficient distinctiveness?  I believe so, but a sentence or two to this effect is required.

The Abstract contains something of a conceptual jump from the past event to a consideration of risk management.  This needs more explanations/elaboration.

Line 78-80.  The point made on different responses to the 1872 event in terms of risk management in Denmark, Germany and Sweden needs some further elaboration as to ‘why’ there were contrasting responses – even if this is reasoned speculation.  I guess this may come later in the paper.

The main body of the text is an interesting elaboration of the event, its impacts and its ramifications for coastal protection measures and strategy – which it brought up to date with a consideration of present policy and approach.

The Discussion is a little underwhelming.  It largely focusses on collective memory of the event in influencing disposition and approach to coastal protection, but surely there is much more to be discussed.  What about the connection of the economy in the areas affected in the different countries to the event?  In any areas where the economy is disconnected to the coast, then there is bound to be limited impact or recognition of vulnerability through the years.  In contrast, those communities whose livelihoods are interwoven with, say, coastal lowland agriculture, as opposed to fishing for example, will remember and be very sensitive to the impact of such events on their health and well-being.  Similarly, what is the role of coastal communities and regions in shaping national policy?  Countries with a strong regional representation in Government would be likely to have a strategy and policy that aligns to protecting these communities and regions.  None of this is explored sufficiently in the Discussion, so the readership is left without sufficient explanation of the ‘why’ in the difference to the response of agencies and government following the 1872 disaster.  From my perspective, I don’t see how a reflective piece such as this would consider only collective memory as the prime influencer.

As a consequence of the missing elements in the Discussion, I believe this is why the Abstract and the Conclusions both seem to have a disconnect between the event-based research and the historical-present  strategy research.  There needs to be a more nuanced elaboration of the ‘why’ rather than just a survey of ‘what was’ and ‘what is’.

See also my comments on the annotated manuscript.

Reviewer 2 Report

I read the manuscript with interest. In my opinion, the end of the manuscript is missing a short section devoted to specific recommendations for raising awareness among residents of coastal lowlands of flood risks and motivation to build reliable protective dikes (walls) and embankments. These recommendations would give your research more practical value.

Other minor recommendations:

  1. It is desirable to combine Figures 2 and 3 into a single figure. The large map of Figure 2 is unnecessary. You need only to write the names of countries, seas, and large islands on the map shown in Figure 3. The map of Figure 2 can be used as an inset map in Figure 3. The final map would look better in color. The latter also applies to Figure 1.
  2. Lines 509-510. What do you mean when you write the following: “However, instead of decreasing the flood risk, these dikes contributed to an increased flood risk since people had relocated to more low-lying areas.”? Please explain in more detail the meaning of this sentence.

Round 2

Reviewer 1 Report

Thank you for making the amendments to the paper.